# Screening for Parkinson's disease using "computer vision"

**Narongrit Kasemsap**[1,2], **Purinat Tikkapanyo**[3], **Panupong Wanjantuk**[4], **Nisa Vorasoot**[1,2☯], **Kannikar Kongbunkiat**[1,2☯], **Anupol Panitchote**[5*]

1 Division of Neurology, Department of Medicine, Faculty of Medicine, Khon Kaen University, Khon Kaen, Thailand, 2 North-Eastern Stroke Research Group, Khon Kaen University, Khon Kaen, Thailand, 3 Department of Medicine, Faculty of Medicine, Khon Kaen University, Khon Kaen, Thailand, 4 Department of Computer Engineering, Faculty of Engineering, Khon Kaen University, Khon Kaen, Thailand, 5 Division of Critical Care Medicine, Department of Medicine, Faculty of Medicine, Khon Kaen University, Khon Kaen, Thailand

☯ These authors contributed equally to this work.
* panupo@kku.ac.th

## Abstract

### Background

Identifying bradykinesia is crucial for diagnosing Parkinson's disease (PD). Traditionally, the finger-tapping test has been used, relying on subjective assessments by physicians. Computer vision offers a non-contact and cost-effective alternative for assessing Parkinson's disease.

### Objective

This study aimed to detect Parkinson's disease by identifying bradykinesia using computer vision in the finger-tapping test and applying machine learning techniques for both hands.

### Methods

We recruited 100 patients with PD and healthy controls. Four neurologists assessed bradykinesia, and 10-second smartphone-recorded finger-tapping movements were analyzed using Google MediaPipe Hands software. Six machine learning models were trained using a nested cross-validation framework.

### Results

The differences in tapping scores between the left and right hands were significantly greater in the PD group (2.8 (5.0) vs 0.4 (0.7), p = 0.001) than in the healthy controls. Moreover, the tapping amplitude variation and all amplitude decremental parameters in the PD group differed significantly from those of the standard controls. The PD group had significantly lower tapping scores than the normal subjects (right: 17.9

**Data availability statement:** Our data are already publicly available. The code and all data underlying the results presented in this study are available on figshare • README. figshare. Dataset. https://doi.org/10.6084/m9.figshare.28883756.v2 • File for setting up the environment. figshare. Dataset. https://doi.org/10.6084/m9.figshare.28931441.v1 • Python code for the Jupyter notebook. figshare. Dataset. https://doi.org/10.6084/m9.figshare.28883624.v2 • Parkinson_data.csv. figshare. Dataset. https://doi.org/10.6084/m9.figshare.28883489.v1 • Example Video : Finger tapping for bradykinesia detection. figshare. Dataset. https://doi.org/10.6084/m9.figshare.28930706.v1 Example extracted data from the finger-tapping task (1 case) for model prediction. figshare. Dataset. https://doi.org/10.6084/m9.figshare.28930694.v1 • SVC model for bradykinesia detection. figshare. Dataset. https://doi.org/10.6084/m9.figshare.28883657.v2.

**Funding:** The study received funding from the Faculty of Medicine, Khon Kaen University, Thailand (Grant Number RU65203) and the Neurological Society of Thailand (no grant number). The respective URLs are https://resmd.kku.ac.th/web/ and http://www.neuroth-ai.org. NK received these grants. The funders had no role in study design, data collection and analysis, decision to publish, or preparation of the manuscript.

**Competing interests:** The authors have declared that no competing interests exist.

(7.8)/ left: 17.9 (5.6) vs. right: 24.6 (7.3)/ left: 24.6 (7.2), p < 0.001). The support vector machine outperformed the other models. The most influential features were the tapping difference, followed by the tapping score (right hand) and tapping amplitude mean (right hand).

## Conclusions

A computer vision method can accurately detect bradykinesia using the tapping feature from the finger-tapping method, which involves the simultaneous tapping of both hands.

---

## Introduction

Parkinson's disease (PD) is clinically defined by the presence of bradykinesia and at least one additional cardinal motor feature (rigidity or rest tremor), along with supporting and exclusionary criteria [1,2]. Accurate identification of bradykinesia is crucial for diagnosing PD, as it is a prerequisite for diagnosis [2]. Various tasks, such as finger tapping, pronation-supination movements, toe-tapping, and foot tapping, have been used subjectively to evaluate bradykinesia. Clinical rating scales, such as the Movement Disorder Society revision of the Unified Parkinson's Disease Rating Scale (MDS-UPDRS), have been developed to assess both motor and non-motor symptoms of PD. These scales have shown high consistency and correlation with the original UPDRS [3]. However, they rely on subjective scoring based on a physician's clinical experience, limiting their ability to detect mild bradykinesia. In order to address this, technology-based tools have been developed to quantify bradykinesia more objectively. These include a computer software program called BRAIN TEST© that assesses finger-tapping speed and accuracy [4]. However, it requires a keyboard and may not be helpful for patients with severe hand dysfunction. Another tool, the Coordination Ability Test System (CATSYS), utilizes sensors such as a tremor pen, touch recording plate, reaction time handle, and force plate for balance recording [5]. While CATSYS correlated with the UPDRS item for pronation/supination, it did not correlate with finger-tapping values and corresponding UPDRS bradykinesia items. Quantitative assessment of finger movement performance on a computer keyboard [6] and musical keyboard [7] demonstrates a correlation with clinical phenomena of repetitive movement and can identify motor signs in early PD [8]. A touchscreen-based tapping application was comparable to a conventional method for assessing motor dysfunction [9]. Gyrosensors have also been used to measure angular movement and have shown a moderate correlation with clinical finger-tap scores [10]. However, these tools require sensors that may not be widely available or specific computer programs with which patients need to interact.

Computer vision (CV) offers several advantages in assessing Parkinson's disease (PD), including its non-contact nature and minimal instrumentation requirements. CV can be easily implemented using standard devices, such as computers with webcams or smartphones with cameras. Previous studies have utilized various computer

vision techniques to track finger movement and normalize tapping signals in PD assessment. These techniques include motion history images and face height detection [11]. In a recent study, smartphone video recording at 60 frames per second (fps) and an optical flow time series were employed for finger detection. The researchers also utilized DeepLabCut, a tool for pose estimation, to quantify bradykinesia. This study demonstrated the potential of computer vision for assessing PD-related bradykinesia [12–14]. The Google MediaPipe Hands (GMH) framework [15], which is open source and compatible with multiple programming languages, including Python, is a powerful tool for extracting hand landmarks from videos. It provides accurate tracking of hand and finger movements, thereby enabling a detailed analysis.

To our knowledge, our study is the first to use computer vision techniques for screening Parkinson's disease by detecting bradykinesia in the finger-tapping test for both hands.

## Materials and methods

The study was approved by the Khon Kaen University Ethics Committee for Human Research (Khon Kaen, Thailand) (Approval #HE644021).

### Participants

We recruited a convenience sample of 53 patients from Srinagarind Hospital, Khon Kaen University. The recruitment period was from February 22, 2022, to November 30, 2023. Participants were selected based on recent visits to the clinic in a hospital setting, without any specific selection criteria. A neurologist had previously diagnosed all patients with Parkinson's disease according to the clinical diagnostic criteria of the Movement Disorder Society [2]. Healthy controls were excluded if they had bradykinesia. Controls without Parkinson's disease or any other neurological diagnosis were recruited from the companions of patients or hospital/university staff members. Patients with motor weakness, joint stiffness, joint deformity, or inability to perform the finger-tapping test due to arthritis or pain were excluded. Written informed consent was obtained from all participants.

Four neurologists rated 100 videos. The raters were blinded to patient or healthy control status and each other's scores. Each video was rated according to the presence of bradykinesia and MDS-UPDRS item 3.4 (finger tapping). For each video, the most common rating given by the clinician was used to measure how well it correlated with a computer vision program assessment. As assessed using Krippendorff's alpha, the overall agreement was 0.72 (95% bootstrap confidence intervals 0.57–0.78) in all cases with at least two ratings. During video recording, the patients were not intentionally withheld from medication.

### Video recording

Finger-tapping movements were recorded using an ordinary smartphone for 10 seconds (s). The recording was performed at a frame rate of 30 fps. The participants were instructed to perform a finger-tapping (FT) test. Before recording, they were asked to stretch their index finger vertically against the thumb as much as possible and then perform tapping as quickly and wide open as possible (i.e., between the thumb and index fingertip as possible). The camera was approximately 0.5–1 m from the hand without a tripod.

### Pose estimation and accuracy

**MediaPipe hands framework.** The Google MediaPipe Hands (GMH) framework is a real-time hand-tracking system that leverages machine learning to detect and predict 21 three-dimensional (3D) landmarks for each hand. The framework consists of two submodules: palm and hand landmark detection. Using a computationally efficient and robust model. This bounding box served as the input for the hand landmark detection submodule. A real-time regression model predicted the 3D coordinates of 21 key landmarks (e.g., fingertips, knuckles) within the detected bounding boxes. The coordinates of

each landmark, including those of the hand and knuckles, consisted of x, y, and z values normalized to the range of 0.0 to 1.0 [16].

The GMH framework has been effectively validated for hand-tracking and motion analysis accuracy. It has been applied to detect and measure resting tremors in patients with Parkinson's disease, and its performance has been confirmed against accelerometer-based systems [17,18]. The validation study compared the GMH with a gold standard system using three dynamic hand exercises, namely hand opening and closing, single-finger tapping, and multiple-finger tapping, which showed strong temporal and spectral consistency. This suggests a distance range of 60–100 cm., various motion speeds (slow, normal, and fast), and a lateral view angle [19].

**Confidence thresholds and handling noisy data.** The GMH framework utilizes a palm detection model as an initial step designed to exclude non-hand objects and reduce the number of false positives. By incorporating bounding box localization and pre-trained models, the framework exhibits robustness across various settings, including challenging conditions such as low light and partial occlusions. These features make it highly applicable to clinical and real-world scenarios, particularly in video-based analysis. The precision of the palm detector has been reported to range between 86.2% and 95.7% [19]. Additionally, the framework ensures stable pose predictions during continuous hand movement by implementing a temporal smoothing mechanism. However, a key limitation is that hand landmarks become unreliable when hands are partially obscured, and detection accuracy declines considerably in environments characterized by significant noise or dynamic activity. For stabilized GMH, temporal filtering techniques stabilize landmark detection across video frames. This helps smooth out the noise and jittering effects that tremors exacerbate. We achieved this by adjusting the confidence thresholds for hand and landmark detection. In this study, we set the min_detection_confidence and min_tracking_confidence to the default value of 0.5.

## Video extraction and pre-processing

The GMH framework [20] was used to extract the finger-tapping videos. This framework provides high-fidelity 3D hand landmarks. The pre-processing stage involved using the OpenCV library (version 4.6.0). We selected 3D data points corresponding to the thumb and index tips. We calculated the Euclidean distance from these data points, which was used as the finger-tapping amplitude. The amplitude values were standardized by dividing all values by the maximum distance in the corresponding time series for each hand. The standardized values were then normalized from 0 to 100, with 100 representing the maximum amplitude. We created time-series data from the finger-tapping amplitude values obtained from both hands of the participants.

## Feature extraction

A peak and valley finder algorithm was used to analyze the tapping signal during the feature extraction process. We employed the find peaks function from Scipy.signal library version 1.7.3 [21], a signal processing package available in Python, to implement this algorithm. A prominence value of 0.3 was set to implement the algorithm. The algorithm identified the peaks (highest points) and valleys (lowest points) in the tapping signal, along with their respective magnitudes and locations. This analysis provided important information about the characteristics and variations of the tapping signal, which could be further used to extract relevant features for subsequent analyses and interpretations.

Finger-tapping (FT) features were derived from the peak amplitudes, which were subsequently used to calculate the features. The tapping signal yielded the following features:

1. The tapping Score (TS) represented the total number of finger taps within a 10-s window.

2. The tapping difference (TD) quantifies the discrepancy in the number of taps between the right and left hand.

3. The mean tapping amplitude (TA) was calculated as the average peak amplitude for each hand.

4. The tapping variation (TV) represents the standard deviation of the peak amplitudes of each hand, indicating the variability of the tapping intensity.

5. Amplitude decrement 5 (AmpD-5) was calculated as the difference between the peak amplitudes of the first and fifth taps.

6. Amplitude decrement 10 (AmpD-10) measures the difference between the peak amplitudes of the first and tenth taps.

An example plot is shown in Fig 1 To visually depict the finger-tapping movements of both hands and their corresponding timestamps. This plot illustrates the finger-tapping activity during the recorded session.

## Model development, optimization, and comparison

We examined a diverse range of machine learning classifiers to identify the one with the highest categorical accuracy for clinical diagnosis of Parkinson's disease, with an MDS-UPDRS part III finger tapping score of at least 1. The Python scikit-learn library (version 1.1.2) [22] was used to train and test the machine learning models using the Jupyter Notebook environment. The study involved the utilization and comparison of six machine learning algorithms: logistic regression (LR), support vector machine (SVM), random forest (RF), decision tree (DT), light gradient boosting model (LGBM), and K-nearest neighborhood (KNN).

We used a nested cross-validation framework to ensure a robust evaluation and prevent information leakage. This approach involved two levels of cross-validation: an outer loop to assess model performance and an inner loop for hyperparameter tuning.

**Outer cross-validation.** The outer loop employed a stratified K-fold cross-validation with 5 splits to ensure that the class distribution of the target variable was preserved across the training and test sets. For each fold, the dataset was split into distinct training and test sets, and the final evaluation metrics were derived from the model performance on the unseen test data.

**Inner cross-validation.** Within each training set generated by the outer loop, a 3-fold stratified K-fold cross-validation was conducted to perform hyperparameter tuning. Various methodologies were explored for tuning the hyperparameters

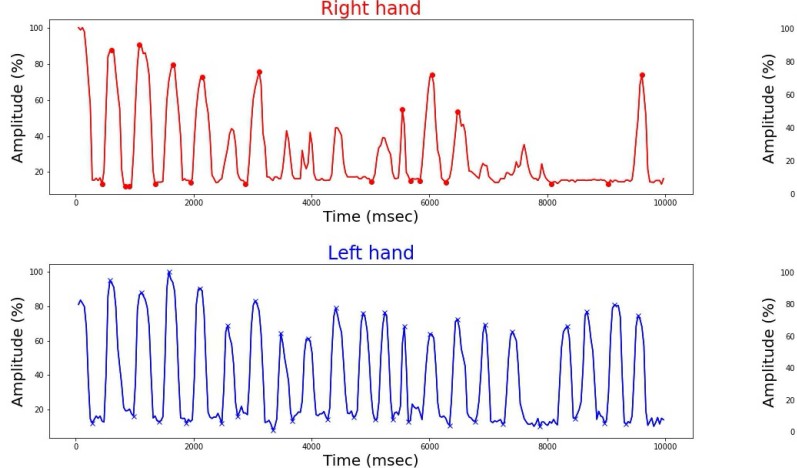
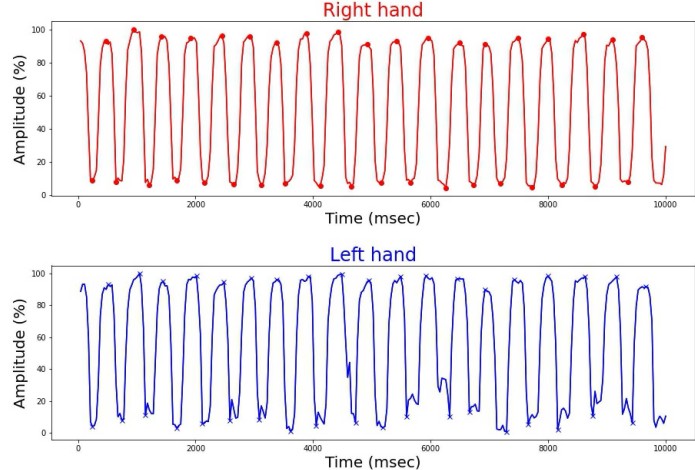

**Fig 1. Finger tapping signals.**

of the six machine learning models using GridSearchCV to predict and analyze the presence of bradykinesia. The hyperparameter configuration that yielded the highest accuracy in the inner loop was selected for training the model on the corresponding outer training set. A similar approach was applied for tuning the hyperparameters of other machine learning algorithms.

**Model evaluation and metrics.** The best model from the inner loop for each outer fold was used to predict the probabilities and class labels on the outer test set. Performance was evaluated using multiple metrics, including accuracy, precision, recall, F1-score, and area under the receiver operating characteristic curve (AUC). Additionally, the AUC was computed to train and test the datasets to assess potential overfitting.

**Confidence interval estimation.** We calculated the mean, standard deviation, and 95% confidence intervals (CIs) for each evaluation metric. The CIs were derived using a t-distribution with degrees of freedom equal to the number of outer folds minus one. These CIs provide insights into the reliability of the reported performance metrics by considering the variability across folds.

This nested cross-validation framework [23], applied across all six machine learning algorithms, ensures that the reported model performance was unbiased, reflective of real-world application, and provides robust insights into the effectiveness of different classifiers.

## Statistical analysis

Numbers and percentages are presented for categorical variables, whereas medians and interquartile ranges (IQRs) were reported for continuous variables that were not normally distributed. Overall agreement across raters was calculated using Krippendorff's alpha [24], which is a helpful measure of overall agreement. Krippendorff's alpha and bootstrap 95% confidence intervals (1,000 iterations, sampling subject with replacement) were calculated using scripts from Zapf A. et al. [25] in R version 4.0.2 (R Foundation for Statistical Computing, Vienna, Austria). The performance of machine learning in predicting the presence of bradykinesia was compared to that of a neurologist using receiver operating characteristics (ROC) analysis, and the area under the curve (AUC) was estimated. The interpretation of AUC was based on the scoring system as follows: 1.0 indicated a perfect test, 0.99–0.90 indicated an excellent test, 0.89–0.8 indicated a good test, 0.79–0.70 indicated a fair test, 0.69–0.51 indicated a poor test and 0.50 or lower indicated a failure [26]. All statistical tests were two-tailed, and $P < 0.05$ was considered significant.

## Results

The study comprised 100 participants:53 patients and 47 healthy control subjects. The baseline characteristics of the patients and healthy controls are presented in Table 1. The two groups exhibited comparable mean ages, with the patients and controls having an average age of $60.6 \pm 10.0$ years and $59.5 \pm 11.7$ years, respectively (p = 0.53). However, there was a significant sex discrepancy (p = 0.003). Videos were recorded in both ON and OFF states. Most patients demonstrated Hoehn and Yahr (H&Y) scale scores of < 3 (83.1%). Furthermore, most of the patients exhibited abnormal finger-tapping to a mild degree (33 hands scored 1, and 40 hands scored 2 in the MDS-UPDRS-III sub-score). In contrast, the PD group had significantly lower tapping scores than the normal subjects (right: 17.9 (7.8)/ left: 17.9 (5.6) vs. right: 24.6 (7.3)/ left: 24.6 (7.2), p<0.001). The differences in tapping scores between the left and right hands were significantly greater in patients with PD than in healthy controls (2.8 (5.0) vs 0.4 (0.7), p = 0.001). Moreover, the tapping amplitude variation and all amplitude decremental parameters in PD subjects differed significantly from those of the standard controls. No parameters were overtly affected by age, sex, or handedness.

Table 2 presents the predictive performance metrics of various machine learning models applied to the testing dataset. The evaluated models include LR, SVM, RF, DT, LGBM, and KNN. The performance metrics considered were accuracy, precision, recall, F1-score, area under the curve (AUC), and training AUC.

**Table 1. Baseline characteristics of the study population.**

| | No bradykinesia (n = 47) | Bradykinesia (n = 53) | | p-value |
|---|---|---|---|---|
| Age, mean (SD), y | 59.5 (11.7) | 60.9 (10.0) | | 0.527 |
| Male, No. (%) | 11 (23.4) | 29 (54.7) | | 0.003 |
| Disease duration, median (IQR), y | – | 2.0 (1.0-5.0) | | – |
| Handedness, right, No. (%) | 39 (83.0) | 42 (89.4) | | 0.550 |
| The most affected side, No. (%) | – | 20/22/3 | | – |
| Right | | 20 (44.4) | | |
| Left | | 22 (48.9) | | |
| Equally | | 3 (6.7) | | |
| ON state, No. (%) | | 22 (46.8) | | – |
| Levodopa use, No. (%) | | 36 (80.0) | | – |
| Hoehn and Yahr Scale, median (IQR) | – | 2.0 (1.0-2.5) | | – |
| 1.0, No. (%) | – | 17 (32.1) | | – |
| 1.5, No. (%) | – | 7 (13.2) | | – |
| 2.0, No. (%) | – | 11 (20.8) | | – |
| 2.5, No. (%) | – | 9 (17.0) | | – |
| 3.0, No. (%) | – | 5 (9.4) | | – |
| 4.0, No. (%) | – | 3 (5.7) | | – |
| 5.0, No. (%) | – | 1 (1.9) | | – |
| | | Right | Left | |
| FT subscore (MDS-UPDRS part III), mean (SD) | – | 1.8 (1.0) | 2.1 (0.7) | – |
| 0, No. (%) | – | 2 (3.8) | 0 (0.0) | – |
| 1, No. (%) | – | 23 (43.4) | 10 (18.9) | – |
| 2, No. (%) | – | 11 (20.8) | 29 (54.7) | – |
| 3, No. (%) | – | 15 (28.3) | 13 (24.5) | – |
| 4, No. (%) | – | 2 (3.8) | 1 (1.9) | – |
| Tapping parameters | | | | |
| Right tapping score, mean (SD) | 24.6 (7.3) | 17.9 (7.8) | | <0.001 |
| Left tapping Score, mean (SD) | 24.6 (7.2) | 17.9 (5.6) | | <0.001 |
| Tapping different, mean (SD) | 0.4 (0.7) | 2.8 (5.0) | | 0.001 |
| Right tapping amplitude, mean (SD), % | 85.4 (6.9) | 71.4 (16.0) | | <0.001 |
| Right tapping amplitude SD, mean (SD), % | 7.9 (3.9) | 14.4 (10.0) | | <0.001 |
| Left tapping amplitude, mean (SD), % | 83.2 (8.0) | 75.3 (10.7) | | <0.001 |
| Left tapping amplitude SD, mean (SD) | 8.0 (4.0) | 13.1 (8.2) | | <0.001 |
| Right amplitude decremental 1–5, mean (SD) | −0.3 (14.1) | 8.6 (18.0) | | 0.007 |
| Right amplitude decremental 1–7, mean (SD) | 0.8 (16.0) | 14.0 (23.1) | | 0.001 |
| Right amplitude decremental 1–10, mean (SD) | 2.6 (15.6) | 20.5 (28.5) | | <0.001 |
| Left amplitude decremental 1–5, mean (SD) | −1.9 (12.8) | 5.2 (17.9) | | 0.024 |
| Left amplitude decremental 1–7, mean (SD) | 0.1 (11.7) | 13.2 (21.6) | | <0.001 |
| Left amplitude decremental 1–10, mean (SD) | 0.4 (11.3) | 17.6 (24.6) | | <0.001 |

FT, finger tapping; SD, standard deviation; MDS-UPDRS, Movement Disorder Society revision of the Unified Parkinson's Disease Rating Scale.

**Table 2. Predictive performance metrics of different models in the testing dataset.**

| Predictive model | Model performance (95%CI) | | | | | Training AUC |
|---|---|---|---|---|---|---|
| | Accuracy[a] | Precision | Recall | F1-score | AUC | |
| Logistic regression | 0.79±0.10 (0.67-0.91) | 0.83±0.15 (0.65-1.02) | 0.79±0.11 (0.66-0.93) | 0.80±0.09 (0.69-0.91) | 0.87±0.07 (0.78-0.97) | 0.96±0.01 (0.95-0.97) |
| Support vector machine | 0.84±0.07 (0.75-0.93) | 0.89±0.10 (0.77-1.02) | 0.81±0.13 (0.65-0.97) | 0.84±0.08 (0.75-0.94) | 0.91±0.05 (0.84-0.98) | 0.96±0.01 (0.95-0.97) |
| Random forest | 0.79±0.08 (0.69-0.89) | 0.82±0.12 (0.67-0.97) | 0.79±0.08 (0.69-0.89) | 0.80±0.08 (0.70-0.89) | 0.87±0.07 (0.77-0.97) | 1.00±0.00 (0.99-1.00) |
| Decision tree | 0.74±0.09 (0.63-0.85) | 0.81±0.14 (0.63-0.98) | 0.72±0.06 (0.64-0.80) | 0.75±0.06 (0.67-0.83) | 0.73±0.09 (0.63-0.85) | 0.98±0.01 (0.97-0.99) |
| Light gradient boosting model | 0.76±0.07 (0.68-0.84) | 0.81±0.13 (0.65-0.97) | 0.75±0.09 (0.64-0.87) | 0.77±0.06 (0.70-0.84) | 0.82±0.09 (0.72-0.93) | 0.99±0.02 (0.97-1.01) |
| K-Nearest Neighbor | 0.76±0.06 (0.69-0.83) | 0.86±0.13 (0.70-1.02) | 0.68±0.05 (0.61-0.74) | 0.75±0.05 (0.69-0.81) | 0.80±0.07 (0.70-0.87) | 0.93±0.06 (0.85-1.00) |

[a]Accuracy was the metric used to evaluate and compare the model performance because accuracy is a suitable metric for a balanced dataset.

Among the models, the SVM achieved the highest accuracy (0.84±0.07, 95% CI: 0.75–0.93), along with substantial precision (0.89±0.10) and recall (0.81±0.13), leading to an F1-score of 0.84±0.08. Additionally, SVM attained the highest AUC in the testing dataset (0.91±0.05), suggesting superior discriminatory ability. The RF and LR models demonstrated comparable performance, with accuracy values of 0.79±0.08 and 0.79±0.10, respectively, and AUC values of 0.87±0.07. The DT model showed the lowest accuracy (0.74±0.09) and AUC (0.73±0.09), indicating lower generalization capability. LGBM and KNN demonstrated moderate performance with accuracy values of 0.76±0.07 and 0.76±0.06, respectively. Notably, the RF model exhibited overfitting, as indicated by a training AUC of 1.00 compared with a lower testing AUC of 0.87. The results suggest that the SVM model provides the best balance between accuracy, precision, recall, and AUC, making it the most suitable choice for this dataset.

Fig 2 shows the ROC curves for the multiple models using nested cross-validation. SVM achieved the highest AUC (0.91±0.05), followed by LR and RF (0.87±0.07), respectively. LGBM and KNN had moderate AUCs of 0.82±0.08 and 0.80±0.07, respectively, whereas the decision tree performed the worst (AUC=0.73±0.09). The dashed line represents a random chance (AUC=0.50). Overall, SVM demonstrated the best discrimination ability among the models.

We employed the SHapley Additive exPlanations (SHAP) method [27] to identify and rank the contributing features in the model's prediction. Shapley values (Fig 3) were used to estimate the importance of the features in the support vector machine model. Each dot represents the influence of tapping features on the prediction. Dots to the left decrease the chance of detecting bradykinesia, whereas dots to the right increase it. The features are listed in descending order of importance, with the tapping difference being the most important, followed by the tapping score (right hand) and tapping amplitude mean (right hand).

## Discussion

The study introduced a novel approach utilizing computer vision techniques with MediaPipe hand to screen for Parkinson's disease by analyzing bradykinesia through finger-tapping tests using smartphone video recordings. Uniquely, our study considered both hands simultaneously, unlike previous studies that often focused on single-hand measurements. This dual-hand assessment enhances the detection of asymmetries and subtle bradykinesia, potentially improving the sensitivity of early Parkinson's disease detection. Notably, our machine learning model, developed from this data, has demonstrated strong predictability regarding bradykinesia, aiming to enhance diagnostic capabilities significantly. The MDS-UPDRS rating employs several crucial features, such as amplitude, speed, and regularity (including hesitations,

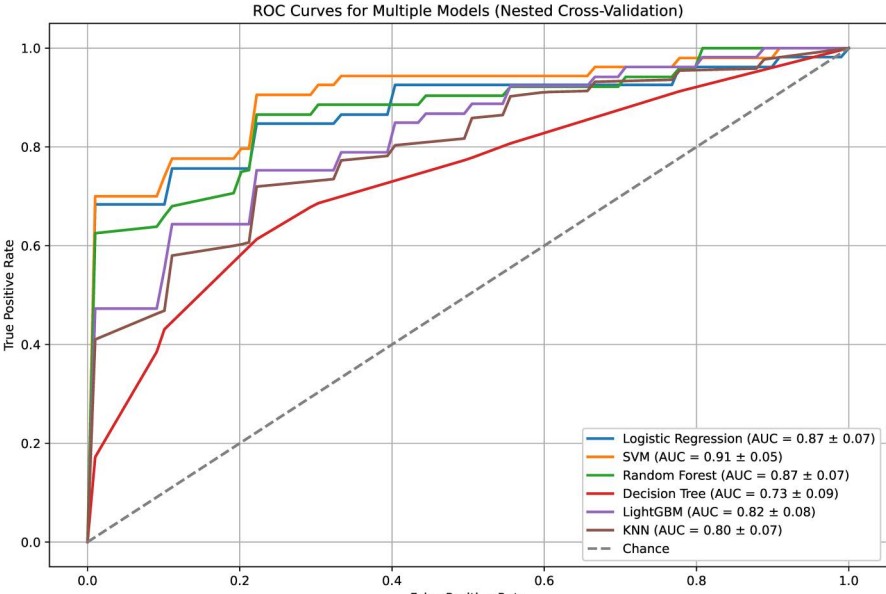

**Fig 2. Receiver operating characteristic (ROC) Curves.**

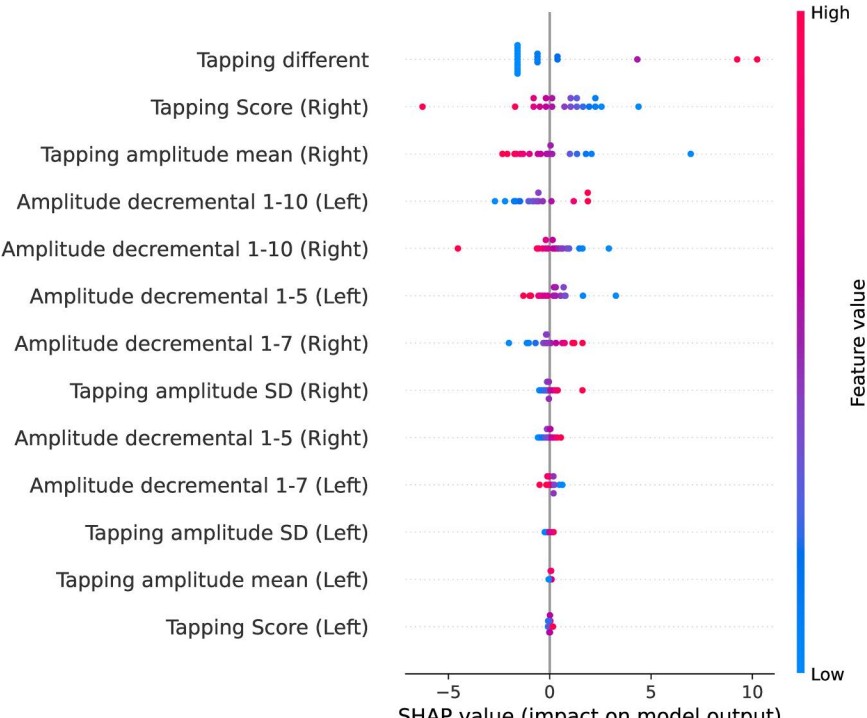

**Fig 3. SHAP summary plot for factors predict bradykinesia.**

halts, and decreasing amplitude). Our study assessed speed (tapping score), regularity (amplitude variation), and amplitude decrement. Previous research utilizing magnetic sensors to extract finger motion properties has identified motion amplitude and the variation of width and frequency as the most significant features [28]. William et al. [12] The peak-to-peak standard deviation, rhythm, and average amplitude were the most predictive measures. Additionally, a study analyzing finger tapping in patients with idiopathic rapid eye movement sleep behavior (RBD) found evidence of amplitude decrement in repetitive movements, a prodromal stage preceding the development of parkinsonism [29]. A study using infrared-emitting diodes found that finger-tapping trials showed slowness with a progressive reduction in amplitude and speed in patients with PD compared with those with PSP and healthy control [30]. This corresponds to our results for the tapping score, average amplitude, amplitude variation, and amplitude decrement. In addition to other studies, PD is an asymmetrical condition; thus, it would be appropriate to assess the performance of the finger-tapping test on both sides. We found significant differences in tapping scores between the two hands in patients with PD. The difference in the number of taps between the right and left hands exhibited the highest SHAP value, underscoring its strong influence on the model's prediction. This metric likely captures asymmetry in finger-tapping performance, a hallmark motor symptom of Parkinson's disease, particularly in the early stages. The second most influential feature, Tapping Score (right), reflects finger-tapping performance, with reduced speed and irregularity being well-established indicators of bradykinesia. These findings highlight the importance of assessing the magnitude and variability of motor actions across the hands. Using computer vision to analyze subtle movement patterns enhances the sensitivity of these measurements, making it particularly valuable for detecting early bradykinesia, where traditional clinical assessments may lack precision. Our results align with those of Yang et al., who used tapping rate and amplitude variation to train a model for assessing the severity of bradykinesia. One key difference is that Yang et al. included freeze time as an additional variable [31]. They then used computer vision to automatically classify bradykinesia through an analysis of finger-tapping performance (viz., the MDS-UPDRS score for multiclass prediction): their deep neural network (DNN) model achieved an average F1-score of 84–88%. Additionally, a convolutional neural network (CNN) model developed by Li et al. achieved an accuracy of 79.7% [31], aiming to classify MDS-UPDRS scores automatically. However, our study focused on screening for Parkinson's disease by detecting bradykinesia and evaluating simultaneous finger tapping, assessing tapping performance in both hands and analyzing inter-hand differences. The study by Wong et al. also included binary classification of the presence of bradykinesia (UPDRS >1) using taping frequency, amplitude, and tap-to-tap variability to predict bradykinesia on the one hand, with an estimated accuracy of 79% [12]. Amprimo et al. utilized the MediaPipe Hand framework and specialized RGB-depth camera (Azure Kinect) to detect bradykinesia, achieving an accuracy of 94.4% and an F1 score of 98.4% in a leave-one-subject-out validation [32]. However, this approach requires a specialized RGB-depth camera for video recording purposes.

There are various publicly available open-source models for pose estimation, such as Openpose [33], DeepLabCut [13,34], AlphaPose [35], YOLOv7 [34], and Mediapipe. Mediapipe can also be utilized on standard hardware, including CPU, GPU, or TPU, and its framework is suitable for a wide range of platforms, such as iOS, Android, desktop, Edge, cloud, web, and IoT. Williams et al. found that DeepLabCut correlated well with a clinical rating for bradykinesia in patients with PD when the video was set at 60 fps [14]. However, the default video frame rates on smartphones are 30 fps. This study used a standard 30 fps, effectively tracking fingertip movement without motion blur. Morinan et al. utilized OpenPose to evaluate five items from the MDS-UPDRS to assess bradykinesia. Their investigation revealed that a binary model (categorizing bradykinesia as low-mild versus moderate-high) performed better in predicting than the five levels of severity: normal, slight, mild, moderate, and severe [36].

The principal benefit of our technique is its ease of deployment, low cost, and contactless nature. Moreover, it can utilize an ordinary smartphone without requiring any specialized settings, thus potentially assisting in early clinical diagnosis. Additionally, preparing video recordings is simple and does not require specific conditions, such as lighting, background, or distance. Our method employs a short recording time that minimizes fatigue. Furthermore, our PD patients did

not require medication discontinuation before performing finger tapping, which is more convenient and relevant to clinical practice. In this study, we analyzed the data on a per-patient basis, in contrast to previous studies that analyzed data on a per-hand basis. We have also developed a web-based platform, pd.kku.ac.th/home, which allows users to screen for Parkinson's disease by uploading a video. The website then processes the video, generates amplitude and time graphs, and provides the results for bradykinesia detection.

The current study had several limitations. First, identifying bradykinesia in patients with subtle symptoms (indicated by finger-tapping subscore of 1) can be challenging. Second, we did not quantitatively measure the absolute distance between the fingertip positions. This omission is noteworthy because the rotational movements of the thumb and finger in the horizontal plane can lead to inaccurate amplitude measurements. Nevertheless, it is worth noting that rotational movements are generally minimal during finger tapping, and we accounted for this by calibrating the maximum tapping distance to 100%. Third, our methodology exclusively focuses on detecting bradykinesia symptoms and may not detect other symptoms of Parkinson's disease. Fourth, our method may encounter difficulties in cases where patients exhibit severe tremors, leading to motion blur that may interfere with accurate measurements. Unfortunately, our study did not include patients with severe tremors. The GMH framework can detect finger movement if the tremor amplitude is less than 10 cm. Finally, although we did not use a tripod for camera setup while recording the videos, most of the videos from patients were able to capture finger movements. Videos with poor recording quality, in which features could not be detected, were excluded. These limitations highlight areas for future research to improve our method and make it more accurate and useful for a wider range of Parkinson's symptoms and patients with Parkinson's disease.

## Conclusions

In summary, in the present study, we demonstrated a computer vision method to detect bradykinesia using the tapping features from the finger-tapping method by simultaneously tapping both hands. The support vector machine algorithm yielded the highest average accuracy (0.84).

## Acknowledgments

The North-Eastern Stroke Research Group, Khon Kaen University, Thailand, supported this study.

We thank Mr. Bryan Roderick Hamman, a native English speaker and member of the Council of Science Editors, for assistance with the English-language presentation of the manuscript under the Khon Kaen University's Publication Clinic, Thailand.

## Author contributions

**Conceptualization:** Narongrit Kasemsap, Purinat Tikkapanyo, Panupong Wanjantuk, Anupol Panitchote.

**Data curation:** Narongrit Kasemsap.

**Formal analysis:** Narongrit Kasemsap.

**Funding acquisition:** Narongrit Kasemsap.

**Methodology:** Narongrit Kasemsap, Panupong Wanjantuk, Anupol Panitchote.

**Project administration:** Purinat Tikkapanyo, Panupong Wanjantuk, Anupol Panitchote.

**Supervision:** Panupong Wanjantuk.

**Validation:** Narongrit Kasemsap.

**Writing – original draft:** Narongrit Kasemsap.

**Writing – review & editing:** Narongrit Kasemsap, Panupong Wanjantuk, Nisa Vorasoot, Kannikar Kongbunkiat, Anupol Panitchote.

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
