## [Decision Letter · Decision Letter 0]

10 Jan 2025

Dear Dr. Panitchote,

Thank you for submitting your manuscript to PLOS ONE. After careful consideration, we feel that it has merit but does not fully meet PLOS ONE’s publication criteria as it currently stands. Therefore, we invite you to submit a revised version of the manuscript that addresses the points raised during the review process.

Two expert reviewers have evaluated the manuscript and despite noted strengths have identified several important concerns that should be addressed in the revised version (Major Revision).

Acceptance of a revised draft will be strongly predicated on the responses to concerns raised by the reviewers:

Please note the recommendations related to a) tempering claims for novelty, b) ensuring adequate representation of current literature and c) clarifying details for the methods. Both Reviewers noted a concern regarding methodological details with respect to executing the pipeline, feature interpretability and model validation.

We look forward to receiving your revised manuscript.

Kind regards,

John A. Thompson

Academic Editor

PLOS ONE

Journal Requirements:

2. Please note that PLOS ONE has specific guidelines on code sharing for submissions in which author-generated code underpins the findings in the manuscript. In these cases, we expect all author-generated code to be made available without restrictions upon publication of the work. Please review our guidelines at https://journals.plos.org/plosone/s/materials-and-software-sharing#loc-sharing-code and ensure that your code is shared in a way that follows best practice and facilitates reproducibility and reuse

Reviewers' comments:

Reviewer's Responses to Questions

**Comments to the Author**

1. Is the manuscript technically sound, and do the data support the conclusions?

Reviewer #1: Yes

Reviewer #2: Yes

2. Has the statistical analysis been performed appropriately and rigorously?

Reviewer #1: Yes

Reviewer #2: Yes

3. Have the authors made all data underlying the findings in their manuscript fully available?

Reviewer #1: No

Reviewer #2: No

4. Is the manuscript presented in an intelligible fashion and written in standard English?

Reviewer #1: Yes

Reviewer #2: No

Reviewer #1: The paper presents a study on detecting bradykinesia in Parkinson disease using computer vision applied to finger-tapping tests. The authors recorded 100 participants performing a 10-second finger-tapping test using a smartphone camera and extracted hand movement data with the Google MediaPipe Hands framework. The statistical analysis and machine learning classification results highlight the potential of non-contact, video-based PD screening. However, there are a few important aspects that can be addressed:

1. Validation of finger-tapping data extraction: the paper describes the use of MediaPipe Hands to estimate 3D landmarks but does not elaborate on how low-confidence landmarks or noisy data were handled. Details on confidence thresholds, data filtering or interpolation would enhance the reliability of the extracted data.

2. Feature interpretability and importance: while the authors use SHAP values to explain feature importance, a deeper exploration and explanation into the clinical significance of the top features would enhance the interpretability of the findings.

3. Handling of severe PD symptoms: The study briefly mentions that severe tremors might cause motion blur, leading to inaccurate measurements. However, no specific strategies to address this issue, such as filtering out affected. Additionally, it would be interesting if the authors check the performances and differences on the severe PD participants.

4. Machine learning validation: the study is limited to a single cohort, which raised concerns about the generalizability of the findings. Testing the model on cross validation or external datasets, or in different clinical settings would strengthen the applicability and reliability of the model.

Reviewer #2: Peer review comments for the manuscript titled “Using computer vision for screening parkinson’ disease”

Thank you for the opportunity to review this manuscript. The study presents a method for utilizing computer vision to diagnose Parkinson’s Disease by analyzing videos of patients engaged in the finger-tapping task. However, there are a few concerns regarding the novel contribution to the field as well as ease-of-use by an interested third party. Below, I outline specific comments and suggestions for the authors to consider:

Major Comments

1. The manuscript claims novelty but does not adequately contextualize the study within the existing body of literature. Several relevant papers are not cited, which makes it difficult to evaluate how this work builds on or diverges from prior studies. Including a thorough literature review and discussing how this work advances the field would help clarify its unique contributions. Specifically, at lines 82:84 the authors state that “To the best of our knowledge, this study represents the first attempt to use computer vision techniques for screening Parkinson's disease by detecting bradykinesia in the finger tapping test for both hands.” However, none of the following four publications were cited, and yet they all describe methods using computer vision to diagnose Parkinson’s Disease by assessing the finger-tapping performance from both hands of their volunteers. To claim novelty, the authors of this manuscript should distinguish their work from these previously published methods.

o Amprimo, G.; Rechichi, I.; Ferraris, C.; Olmo, G. Objective Assessment of the Finger Tapping Task in Parkinson’s Disease and Control Subjects using Azure Kinect and Machine Learning. In Proceedings of the 2023 IEEE 36th International Symposium on Computer-Based Medical Systems (CBMS), L’Aquila, Italy, 22–24 June 2023; pp. 640–645.

o Wong, D.C.; Relton, S.D.; Fang, H.; Qhawaji, R.; Graham, C.D.; Alty, J.; Williams, S. Supervised Classification of Bradykinesia for Parkinson’s Disease Diagnosis from Smartphone Videos. In Proceedings of the 2019 IEEE 32nd International Symposium on Computer-Based Medical Systems (CBMS), Cordoba, Spain, 5–7 June 2019; pp. 32–37.

o Li, Z.; Lu, K.; Cai, M.; Liu, X.; Wang, Y.; Yang, J. An Automatic Evaluation Method for Parkinson’s Dyskinesia Using Finger Tapping Video for Small Samples. J. Med. Biol. Eng. 2022, 42, 351–363.

o Yang, N.; Liu, D.F.; Liu, T.; Han, T.; Zhang, P.; Xu, X.; Lou, S.; Liu, H.G.; Yang, A.C.; Dong, C.; et al. Automatic Detection Pipeline for Accessing the Motor Severity of Parkinson’s Disease in Finger Tapping and Postural Stability. IEEE Access 2022, 10, 66961–66973.

2. Additionally, the "novel contribution" of the study is not clearly articulated. It would be beneficial for the authors to explicitly state what distinguishes their approach or findings from existing work.

3. The authors highlight that their method is easy to use, which is a commendable goal. However, the manuscript does not provide sufficient information on where to find the necessary resources (e.g., code, tools, datasets) or detailed instructions on how to use them. Including a link to the resources, accompanied by a brief user guide or example, would significantly enhance the usability and reproducibility of the method.

Minor Comments

1. Line 58 – The authors should briefly state why BRAIN TEST is inferior to their method.

2. Line 100 – For clarity, please rework the sentence: “The modal clinician rating for each video was used for correlation with computer vision programmed.”

3. I sympathize with the challenge of scientific writing in a non-native language, however the manuscript would benefit from careful editing, as multiple grammatical errors and typos were identifiable.

Suggestions for Improvement

• Include a comprehensive discussion of related work, with appropriate citations.

• Clearly articulate the novel contribution of the study in the introduction and discussion sections.

• Provide explicit guidance on accessing and using the method, such as a supplementary document or a dedicated section in the manuscript.

• Add more detailed explanations in the methods section to ensure reproducibility.

• Revise and refine the manuscript’s language for clarity and precision.

I appreciate the effort the authors have put into this manuscript and encourage them to address these points to maximize its impact. I look forward to seeing a revised version.

**Do you want your identity to be public for this peer review?** For information about this choice, including consent withdrawal, please see our Privacy Policy

Reviewer #1: No

Reviewer #2: **Yes: ** W. Ryan Williamson

---

## [Author Response · Author response to Decision Letter 1]

1 Mar 2025

TO:

John A. Thompson

Academic Editor

PLOS ONE

Re: PLOS ONE

PONE-D-24-40785

Using computer vision for screening parkinson’ disease

Dear editors

We are very grateful for the constructive reviewers’ comments, for the opportunity to improve upon our original manuscript, and for your consideration of this revised version. We have made changes to the original manuscript in order to address the reviewers’ concerns. These changes have been listed below. We have shared the code and data, which can be accessed via the URL provided in the Data availability section. All data has already de-identified.

We sincerely believe that the changes suggested by the reviewers have strengthened our study and will make for a greater impact of this article on the field of Parkinson’ disease. Thank you very much for your consideration of this updated version.

Sincerely,

Anupol Panitchote, MD.

(on behalf of all the authors)

Response to reviewer comment

Reviewer #1: The paper presents a study on detecting bradykinesia in Parkinson disease using computer vision applied to finger-tapping tests. The authors recorded 100 participants performing a 10-second finger-tapping test using a smartphone camera and extracted hand movement data with the Google MediaPipe Hands framework. The statistical analysis and machine learning classification results highlight the potential of non-contact, video-based PD screening. However, there are a few important aspects that can be addressed:

1. Validation of finger-tapping data extraction: the paper describes the use of MediaPipe Hands to estimate 3D landmarks but does not elaborate on how low-confidence landmarks or noisy data were handled. Details on confidence thresholds, data filtering or interpolation would enhance the reliability of the extracted data.

Answer: We add this paragraph to the Material and methods (Pose estimation and accuracy) part.

“MediaPipe Hands framework

The Google MediaPipe Hands (GMH) framework is a real-time hand-tracking system that leverages machine learning to detect and predict 21 three-dimensional (3D) landmarks for each hand. The framework consists of two submodules: palm detection and hand landmark detection. The palm detection submodule identifies a bounding box around the hand using a computationally efficient and robust model. This bounding box serves as the input for the hand landmark detection submodule. Within the detected bounding box, a regression model predicts the 3D coordinates of 21 key landmarks (e.g., fingertips, knuckles) in real time. The coordinates of each landmark, including those of the hand knuckles, consist of x, y, and z values, normalized to the range of 0.0 to 1.0 [16].

The GMH framework has been effectively validated for its accuracy in hand tracking and motion analysis. It has been applied in detecting and measuring resting tremors in Parkinson’s disease, with its performance confirmed against accelerometer-based systems The Google MediaPipe Hand (GMH) is a deep learning library utilized for hand tracking in a light-weight and portable machine learning pipeline. It consists of two sub-modules: palm detection and hand landmarks detection. GMH has been applied in detecting and measuring resting tremor in Parkinson’s disease, with its accuracy confirmed against an accelerometer [16, 17]. The validation study compares the GMH with a gold standard system using three dynamic hand exercises, namely hand open-closing, single finger tapping and multiple finger tapping, showing strong temporal and spectral consistency. It suggests a distance range of 60-100 cm, various motion speed (slow, normal, fast), and a lateral view angle [18].

Confidence Thresholds and Handling Noisy Data

The GMH framework utilizes a palm detection model as an initial step, specifically designed to exclude non-hand objects and reduce false positives. By incorporating bounding box localization and pre-trained models, the framework exhibits robustness across a variety of settings, including challenging conditions such as low light and partial occlusions. These features make it highly applicable to clinical and real-world scenarios, particularly in video-based motion analysis. The precision of the palm detector has been reported to range between 86.2% and 95.7% [19]. Additionally, the framework ensures stable pose predictions during continuous hand movements through the implementation of a temporal smoothing mechanism. However, a key limitation is that hand landmarks become unreliable when hands are partially obscured, and detection accuracy declines considerably in environments characterized by significant noise or dynamic activity. For stablizied GMH employs temporal filtering techniques to stabilize landmark detection across video frames. This helps to smooth out noise and jittering effects that might be exacerbated by tremors. We achieve this by adjusting the confidence thresholds used for both hand detection and landmark detection. In this study, we set the min_detection_confidence and min_tracking_confidence to the default value of 0.5.”

2. Feature interpretability and importance: while the authors use SHAP values to explain feature importance, a deeper exploration and explanation into the clinical significance of the top features would enhance the interpretability of the findings.

Answer: We add this paragraph to the discussion part.

“The difference in the number of taps between the right and left hands exhibits the highest SHAP value, underscoring its strong influence on the model's prediction. This metric likely captures asymmetry in finger-tapping performance, a hallmark motor symptom of Parkinson’s disease, particularly in its early stages. The second most influential feature, Tapping Score (Right), reflects finger-tapping performance, with reduced speed and irregularity being well-established indicators of bradykinesia. The third, Amplitude Decrement 1-10 (Left), measures the progressive decline in tapping amplitude over repeated tasks, a key characteristic of bradykinesia. These findings highlight the importance of assessing both the magnitude and variability of motor actions across hands. The use of computer vision to analyze subtle movement patterns enhances the sensitivity of these measurements, making it particularly valuable for detecting early bradykinesia, where traditional clinical assessments may lack precision. These results align with the study by Yang et al., which utilized three features—tapping rate and tapping amplitude variation—to train a classification model for assessing bradykinesia severity. However, one key difference is that their study included tapping freeze time as an additional variable.”

3. Handling of severe PD symptoms: The study briefly mentions that severe tremors might cause motion blur, leading to inaccurate measurements. However, no specific strategies to address this issue, such as filtering out affected. Additionally, it would be interesting if the authors check the performances and differences on the severe PD participants.

Answer: We add this paragraph to the material and methods (Confidence and Thresholds and handling noisy data) part.

…. For stablizied GMH employs temporal filtering techniques to stabilize landmark detection across video frames. This helps to smooth out noise and jittering effects that might be exacerbated by tremors. We achieve this by adjusting the confidence thresholds used for both hand detection and landmark detection. In this study, we set the min_detection_confidence and min_tracking_confidence to the default value of 0.5.

We add this paragraph to the Discussion part.

… Unfortunately, in our study, we didn't have patients with severe tremors; we tested on normal subjects. The GMH framework can detect tremors with an amplitude of less than 10 cm.

4. Machine learning validation: the study is limited to a single cohort, which raised concerns about the generalizability of the findings. Testing the model on cross validation or external datasets, or in different clinical settings would strengthen the applicability and reliability of the model.

Answer: We add this paragraph to the Material and methods part

Model development, validation and performance evaluation

We examined a diverse range of machine learning classifiers to identify the one with the highest categorical accuracy for the clinical diagnosis of Parkinson’s disease, specifically for patients with an MDS-UPDRS part III finger-tapping score of at least 1. The Python scikit-learn library, version 1.1.2[21], was utilized to train and test machine learning models in a Jupyter Notebook environment. The study involved the utilization and comparison of six machine learning algorithms, namely logistic regression classifier (LR), support vector machine classifier (SVM), random forest classifier (RF), decision tree (DT), light gradient boosting model (LGBM), and K-nearest neighbor classifier (KNN).

To ensure robust evaluation and prevent information leakage, we utilized a nested cross-validation framework. This approach involved two levels of cross-validation: an outer loop to assess model performance and an inner loop for hyperparameter tuning.

Outer Cross-Validation:

The outer loop employed a Stratified K-Fold Cross-Validation with 5 splits to ensure the class distribution of the target variable was preserved across training and test sets. For each fold, the dataset was split into distinct training and test sets, and the final evaluation metrics were derived from the model's performance on the unseen test data.

Inner Cross-Validation:

Within each training set generated by the outer loop, a 3-fold Stratified K-Fold Cross-Validation was conducted to perform hyperparameter tuning. Various methodologies were explored for tuning the hyperparameters of the six machine learning models, using GridSearchCV, to predict and analyze the presence of bradykinesia. The hyperparameter configuration yielding the highest accuracy in the inner loop was selected for training the model on the corresponding outer training set. A similar approach was applied for tuning the hyperparameters of other machine learning algorithms.

Model Evaluation and Metrics:

For each outer fold, the best model from the inner loop was used to predict probabilities and class labels on the outer test set. Performance was evaluated using multiple metrics, including accuracy, precision, recall, F1-score, and the area under the receiver operating characteristic curve (AUC). Additionally, the AUC was computed for both training and testing datasets to assess potential overfitting.

Confidence Interval Estimation:

To provide an estimate of the variability of the model's performance across folds, we calculated the mean, standard deviation, and 95% confidence intervals (CIs) for each evaluation metric. The CIs were derived using a t-distribution with degrees of freedom equal to the number of outer folds minus one. These CIs provide insight into the reliability of the reported performance metrics, considering variability across folds.

This nested cross-validation framework [23] applied across all six machine learning algorithms, ensures that the reported model performance is unbiased, reflective of real-world application, and provides robust insight into the effectiveness of different classifiers.

Reviewer #2: Peer review comments for the manuscript titled “Using computer vision for screening parkinson’ disease”

Thank you for the opportunity to review this manuscript. The study presents a method for utilizing computer vision to diagnose Parkinson’s Disease by analyzing videos of patients engaged in the finger-tapping task. However, there are a few concerns regarding the novel contribution to the field as well as ease-of-use by an interested third party. Below, I outline specific comments and suggestions for the authors to consider:

Major Comments

1. The manuscript claims novelty but does not adequately contextualize the study within the existing body of literature. Several relevant papers are not cited, which makes it difficult to evaluate how this work builds on or diverges from prior studies. Including a thorough literature review and discussing how this work advances the field would help clarify its unique contributions. Specifically, at lines 82:84 the authors state that “To the best of our knowledge, this study represents the first attempt to use computer vision techniques for screening Parkinson's disease by detecting bradykinesia in the finger tapping test for both hands.” However, none of the following four publications were cited, and yet they all describe methods using computer vision to diagnose Parkinson’s Disease by assessing the finger-tapping performance from both hands of their volunteers. To claim novelty, the authors of this manuscript should distinguish their work from these previously published methods.

o Amprimo, G.; Rechichi, I.; Ferraris, C.; Olmo, G. Objective Assessment of the Finger Tapping Task in Parkinson’s Disease and Control Subjects using Azure Kinect and Machine Learning. In Proceedings of the 2023 IEEE 36th International Symposium on Computer-Based Medical Systems (CBMS), L’Aquila, Italy, 22–24 June 2023; pp. 640–645.

o Wong, D.C.; Relton, S.D.; Fang, H.; Qhawaji, R.; Graham, C.D.; Alty, J.; Williams, S. Supervised Classification of Bradykinesia for Parkinson’s Disease Diagnosis from Smartphone Videos. In Proceedings of the 2019 IEEE 32nd International Symposium on Computer-Based Medical Systems (CBMS), Cordoba, Spain, 5–7 June 2019; pp. 32–37.

o Li, Z.; Lu, K.; Cai, M.; Liu, X.; Wang, Y.; Yang, J. An Automatic Evaluation Method for Parkinson’s Dyskinesia Using Finger Tapping Video for Small Samples. J. Med. Biol. Eng. 2022, 42, 351–363.

o Yang, N.; Liu, D.F.; Liu, T.; Han, T.; Zhang, P.; Xu, X.; Lou, S.; Liu, H.G.; Yang, A.C.; Dong, C.; et al. Automatic Detection Pipeline for Accessing the Motor Severity of Parkinson’s Disease in Finger Tapping and Postural Stability. IEEE Access 2022, 10, 66961–66973.

Answer: We add this paragraph to the discussion part.

The study employs computer vision to automatically classify bradykinesia by analyzing finger-tapping performance using the MDS-UPDRS score for multiclass prediction. A deep neural network (DNN) model achieved an average F1-score of 84–88%, as reported by Yang et al. Additionally, a convolutional neural network (CNN) model developed by Li et al. achieved an accuracy of 79.7% [31], aiming to automatically classify MDS-UPDRS scores. However, our study focuses on screening Parkinson’s disease by detecting bradykinesia and evaluating simultaneous finger tapping, assessing tapping performance in both hands, and analyzing inter-hand differences.The study by DC Wong, et al. also perform binary classficiation of presence of bradykinesia (UPDRS >1) using taping frequency, amplitude and tap-to-tap variability to predict bradykinesia on one hands, have a estimate accuracy 79% [32]. Amprimo et al. utilized the MediaPipe Hand framework and a specialized RGB-depth camera (Azure Kinect) to detect bradykinesia, achieving an accuracy of 94.4% and an F1 score of 98.4% in a leave-one-subject-out validation [33]. However, this approach requires a specialized RGB-depth camera for video recording.

2. Additionally, the "novel contribution" of the study is not clearly articulated. It would be beneficial for the authors to explicitly state what distinguishes their approach or findings from existing work.

Answers: We add this paragraph to the discussion part.

In this study, we introduced a novel approach utilizing computer vision techniques with MediaPipe hand to screen for Parkinson's disease by analyzing bradykinesia through finger tapping tests on smartphone video recordings. Uniquely, our study considers both hands simultaneously, contrasting with previous research that often focuses on a single hand measures. This dual-hand assessment enhances the detection of asymmetries and subtle bradykinesia, potentially improving early Parkinson's disease detection sensitivity. Notably, our machine learning model, developed from this data, has demonstrated strong predictability regarding bradykinesia, aiming to enhance diagnostic capabilities significantly.

3. The authors highlight that their method is easy to use, which is a commendable goal. However, the m

---

## [Decision Letter · Decision Letter 1]

28 Apr 2025

Dear Dr. Panitchote,

Thank you for submitting your manuscript to PLOS ONE. After careful consideration, we feel that it has merit but does not fully meet PLOS ONE’s publication criteria as it currently stands. Therefore, we invite you to submit a revised version of the manuscript that addresses the points raised during the review process.

Thank you for submitting the revised version of your manuscript, “Using computer vision for screening parkinson’ disease,” to PLOS ONE. Your work was re-evaluated by the two original reviewers. Reviewer 1 now recommends Accept, noting that their concerns were fully addressed. Reviewer 2 sees merit in the study but still identifies critical issues that prevent publication at this time.

After carefully considering both reviews and the journal’s publication criteria, I am issuing a Major Revision decision. The core scientific conclusions appear valid; however, two outstanding concerns remain unmet with regard to Reviewer 2's concerns:

1. Clarity of reporting

Several sentences remain difficult to parse, obscuring key ideas. Please see comment raised by Reviewer 2 regarding the issue of clarity.

2. Reproducibility and open materials

PLOS ONE requires that “all data and author-generated code necessary to replicate the study’s findings be publicly available at the time of publication”.

o Expand the Jupyter notebook into a self-contained, step-by-step workflow (environment setup, data download, execution order).

o Provide a requirements.txt or environment.yml.

o Move all raw and intermediate data needed to run the notebook into a stable repository (e.g., Zenodo, Figshare, OSF) and cite the DOI in the Data-Availability Statement.

o Supply a brief README describing folder structure and expected outputs.

3. Functioning online tool

Reviewer 2 could not access the web interface. Please ensure the site is live for anonymous testing, or remove the claim from the manuscript.

If you believe any request is unreasonable, please justify this in your rebuttal.

We appreciate your commitment to transparent and reproducible science and look forward to receiving your resubmission.

We look forward to receiving your revised manuscript.

Kind regards,

John A. Thompson

Academic Editor

PLOS ONE

Reviewers' comments:

Reviewer's Responses to Questions

**Comments to the Author**

Reviewer #1: All comments have been addressed

Reviewer #2: (No Response)

2. Is the manuscript technically sound, and do the data support the conclusions?

Reviewer #1: Yes

Reviewer #2: Partly

3. Has the statistical analysis been performed appropriately and rigorously?

Reviewer #1: N/A

Reviewer #2: I Don't Know

4. Have the authors made all data underlying the findings in their manuscript fully available?

Reviewer #1: No

Reviewer #2: No

5. Is the manuscript presented in an intelligible fashion and written in standard English?

Reviewer #1: Yes

Reviewer #2: No

Reviewer #1: The authors have addressed the previous concerns raised in the first review round. The expanded explanation of MediaPipe Hands, implementation of SHAP interpretation, and description of the nested cross-validation framework are appreciated.

Reviewer #2: I appreciate the authors’ time and effort in revising the manuscript and providing detailed responses to the initial round of feedback. However, after reviewing the revised version and accompanying materials, I believe that several critical issues remain unresolved, and I must recommend rejection at this time.

1. Clarity of Writing

Unfortunately, the clarity of the manuscript remains a significant concern. Many sentences are difficult to interpret, and in several instances, the intended meaning is unclear. While it is entirely understandable that English may not be the authors’ first language, the current level of writing poses a barrier to understanding the core ideas and contributions of the work. I encourage the authors to seek assistance from a fluent English speaker or professional editing service to improve the readability of the manuscript before resubmission elsewhere.

2. Code Availability and Reproducibility

In response to requests for code sharing, the authors provided a Jupyter notebook and a rudimentary readme document. However, the notebook is not sufficiently structured for others to understand or reproduce the analysis. A well-organized and clearly annotated codebase would be more appropriate and aligned with the journal’s expectations for open and reproducible science.

3. Functionality of Online Tool

The authors mention the development of an online resource intended to make their work more accessible. However, at the time of review, the website does not appear to function as intended. I encourage the authors to ensure that any publicly accessible resources are fully operational before including them in future submissions.

While the study may contain interesting ideas, the combination of unclear writing, inadequate code availability, and non-functioning supplemental tools currently prevents the manuscript from meeting the standards for publication in PLOS ONE. I hope the authors will find this feedback helpful as they work to further develop and refine their work for future submission.

**Do you want your identity to be public for this peer review?** For information about this choice, including consent withdrawal, please see our Privacy Policy

Reviewer #1: No

Reviewer #2: No

---

## [Author Response · Author response to Decision Letter 2]

28 May 2025

TO:

John A. Thompson

Academic Editor

PLOS ONE

Re: PLOS ONE

PONE-D-24-40785

Screening for Parkinson’s disease using “computer vision”

Dear editors

We have appreciation to the reviewers for their thoughtful and constructive feedback, which has greatly contributed to improving the quality of our manuscript. We have carefully addressed all comments, and the corresponding revisions are outlined below.

To improve the clarity and precision of reporting, the revised manuscript has undergone thorough language editing by a native English speaker. Moreover, transparency and reproducibility of our work, we have shared the complete de-identified dataset and annotated analysis code on Figshare, accessible via the DOI provided in the Data Availability section. Furthermore, we have developed a user-friendly online tool that allows others to interactively explore the model and apply it to their own data. We believe this open-access approach will facilitate broader use and validation of our findings.

We are confident that these improvements have strengthened the manuscript and increased its potential impact within the field.

Sincerely,

Anupol Panitchote, MD.

(on behalf of all the authors)

Review Comments to the Author

Reviewer #1: The authors have addressed the previous concerns raised in the first review round. The expanded explanation of MediaPipe Hands, implementation of SHAP interpretation, and description of the nested cross-validation framework are appreciated.

Reviewer #2: I appreciate the authors’ time and effort in revising the manuscript and providing detailed responses to the initial round of feedback. However, after reviewing the revised version and accompanying materials, I believe that several critical issues remain unresolved, and I must recommend rejection at this time.

1. Clarity of Writing

Unfortunately, the clarity of the manuscript remains a significant concern. Many sentences are difficult to interpret, and in several instances, the intended meaning is unclear. While it is entirely understandable that English may not be the authors’ first language, the current level of writing poses a barrier to understanding the core ideas and contributions of the work. I encourage the authors to seek assistance from a fluent English speaker or professional editing service to improve the readability of the manuscript before resubmission elsewhere.

Answer: We sincerely appreciate the reviewer’s feedback regarding the clarity of the manuscript. To address this concern, we have sought assistance from Mr. Bryan Roderick Hamman, a native English speaker and a member of the Council of Science Editors. He provided support with the English-language presentation of the manuscript through Khon Kaen University’s Publication Clinic, Thailand. We believe the revised version is now significantly clearer and more readable.

2. Code Availability and Reproducibility

In response to requests for code sharing, the authors provided a Jupyter notebook and a rudimentary readme document. However, the notebook is not sufficiently structured for others to understand or reproduce the analysis. A well-organized and clearly annotated codebase would be more appropriate and aligned with the journal’s expectations for open and reproducible science.

Answer: We have improved the structure and clarity of our shared resources to better align with the journal’s expectations for open and reproducible science. Specifically:

• We now provide a detailed README file outlining the step-by-step workflow.

• A Jupyter Notebook with clearer annotations and logical structure is included.

• An environment.yml file is provided to facilitate the setup of the required computational environment.

All code, data, and supporting files necessary to reproduce our results are publicly available on Figshare at the following links:

• README: https://doi.org/10.6084/m9.figshare.28883756.v2

• Environment setup file: https://doi.org/10.6084/m9.figshare.28931441.v1

• Jupyter Notebook: https://doi.org/10.6084/m9.figshare.28883624.v2

• Dataset (Parkinson_data.csv): https://doi.org/10.6084/m9.figshare.28883489.v1

• Example video (finger tapping): https://doi.org/10.6084/m9.figshare.28930706.v1

• Example extracted data (1 case): https://doi.org/10.6084/m9.figshare.28930694.v1

• SVC model file: https://doi.org/10.6084/m9.figshare.28883657.v2

We hope these revisions adequately address your concerns and make the study fully reproducible.

3. Functionality of Online Tool

The authors mention the development of an online resource intended to make their work more accessible. However, at the time of review, the website does not appear to function as intended. I encourage the authors to ensure that any publicly accessible resources are fully operational before including them in future submissions.

Answer: We have identified and resolved the issue that affected the website’s performance during the review period. The online tool is now fully operational and accessible at: KKU Parkinson Detection (https://pd.kku.ac.th/home). We have verified that all core functionalities are working as intended, and we will continue to monitor and maintain the platform to ensure its reliability.

---

## [Decision Letter · Decision Letter 2]

5 Jun 2025

Dear Dr. Panitchote,

Thank you for submitting your manuscript to PLOS ONE. After careful consideration, we feel that it has merit but does not fully meet PLOS ONE’s publication criteria as it currently stands. Therefore, we invite you to submit a revised version of the manuscript that addresses the points raised during the review process.

We look forward to receiving your revised manuscript.

Kind regards,

John A. Thompson

Academic Editor

PLOS ONE

Journal Requirements:

Reviewers' comments:

Reviewer's Responses to Questions

**Comments to the Author**

Reviewer #2: (No Response)

2. Is the manuscript technically sound, and do the data support the conclusions?

Reviewer #2: Yes

3. Has the statistical analysis been performed appropriately and rigorously?

Reviewer #2: Yes

4. Have the authors made all data underlying the findings in their manuscript fully available?

Reviewer #2: Yes

5. Is the manuscript presented in an intelligible fashion and written in standard English?

Reviewer #2: Yes

Reviewer #2: The authors should be commended for the effort to create a website that detects bradykinesia in a video uploaded from mobile devices. However, there are a couple of problems that need to be addressed.

1) The website only works if all fingers are visible in every frame. If an individual clicks record, sets the phone down, and then starts tapping, the video is not usable and hangs without warning or explanation. The authors should either add more detail to the instructions or, preferably, edit the software to ignore early frames in which both hands may not be visible. If an error does occur, the website should alert the user of the error instead of hanging indefinitely.

2) Considering the popularity of the iPhone and other Apple products, the authors should include .mov as an acceptable video file, since .mov is the native file extension on Apple products.

3) According to your Materials and Methods section, the healthy participants in your study were excluded if they had bradykinesia, motor weakness, joint stiffness, joint deformity, or inability to perform the finger-tapping test due to arthritis or pain. The website should alert users that a false-positive diagnosis could result if they have any of these conditions.

The authors should be commended for their effort editing the manuscript for clarity. The following list includes the remaining errors which need to be corrected.

line 38 "features was" should be "features were"

line 57 "patient" should be "patients"

line 76 "forassessing" should be "for assessing"

line 93 "perfrom" should be "perform"

line 95 "Four neurologists were rate 100 videos" should be "Four neurologists rated 100 videos"

line 133 "implementating" should be "implementing"

line 136 "stablizied" should be "stabilized"

line 142 insert a space before the bracket

line 177 insert a space after the comma

line 178 insert a space before parenthesis

line 179 "enviroment" should be "environment"

line 160 The text should be "within a 10-s window"

line 196 "for train the model" should be "for training the model"

line 197 "tune" should be "tuning"

line 214 insert a space before the bracket

line 222 "fail" should be "failure"

line 228 insert a space before the parenthesis

line 231 "slightmild" should be either "slight" or "mild"

line 254 "choice for for this" should be "choice for this"

line 257 replace "for the" with "to"

line 266 insert a space after "values"

line 281 insert a space after the period

line 285 insert a space before "[12]"

line 295 insert a space after the period

line 312 "classficiation" should be "classification"

line 319 "are various of" should be "are various"

line 329 "are" should be "is"

line 330 "it is can utilize" should be "it can utilize"

line 349 "tremors amplitude are" should be "tremor amplitude is"

line 352 insert a space after the period

line 366 "supportedby" should be "supported by"

In general, there should be a space before all brackets and parentheses, and there should be a space after all commas.

**Do you want your identity to be public for this peer review?** For information about this choice, including consent withdrawal, please see our Privacy Policy

Reviewer #2: **Yes: ** W. Ryan Williamson

---

## [Author Response · Author response to Decision Letter 3]

4 Jul 2025

TO:

John A. Thompson

Academic Editor

PLOS ONE

Re: PLOS ONE

PONE-D-24-40785

Screening for Parkinson’s disease using “computer vision”

Dear editors

We have appreciation to the reviewers for their thoughtful and constructive feedback, which has greatly contributed to improving the quality of our manuscript. We have carefully addressed all the comments, and the corresponding revisions are outlined below. We have also carefully checked the English grammar in response to the reviewers’ suggestions. Furthermore, we have corrected the website and revised some of the code, as requested.

For the references, we did not cite the retracted publication, and we have also updated Reference 12 with the final published version of the article.

We are confident that these improvements have strengthened the manuscript and increased its potential impact within the field.

Sincerely,

Anupol Panitchote, MD.

(on behalf of all the authors)

Reviewer #2: The authors should be commended for the effort to create a website that detects bradykinesia in a video uploaded from mobile devices. However, there are a couple of problems that need to be addressed.

1) The website only works if all fingers are visible in every frame. If an individual clicks record, sets the phone down, and then starts tapping, the video is not usable and hangs without warning or explanation. The authors should either add more detail to the instructions or, preferably, edit the software to ignore early frames in which both hands may not be visible. If an error does occur, the website should alert the user of the error instead of hanging indefinitely.

Answer: We have addressed this by modifying the code to be more robust. The application will now automatically detect when finger tapping begins within the video. It will then proceed to process the video and generate the graph specifically from the point where the finger tapping action is identified. This ensures that only the relevant segments of the video are analyzed, and it will also ensure a total duration of at least 10 seconds of actual tapping is used for analysis. (This addresses the scenario where a user might start recording, setting the phone down, and then begin tapping).

The images provided illustrate this improvement:

Original scenario (left graphs): Shows erratic data before the tapping truly begins.

Improved scenario (right graphs): Demonstrates that the analysis now accurately starts from the moment finger tapping is detected, providing cleaner and more reliable data, ensuring 10 seconds of tapping is analyzed.

This adjustment will prevent the website from hanging and will provide a more seamless and user-friendly experience, as it no longer requires users to ensure fingers are visible from the very beginning of the recording.

After downloading the .mov file, the preview is currently not supported. However, the website is able to detect the .mov file correctly. We are working on resolving this issue. The graph, however, is displaying properly.

2) Considering the popularity of the iPhone and other Apple products, the authors should include .mov as an acceptable video file, since .mov is the native file extension on Apple products.

Answer: We have updated our website and application requirements to include .mov as an acceptable video file format. The revised text is as follows: “Please upload your video to the application. It must be in .mp4 or .mov format and recorded at 30 frames per second (fps).”

3) According to your Materials and Methods section, the healthy participants in your study were excluded if they had bradykinesia, motor weakness, joint stiffness, joint deformity, or inability to perform the finger-tapping test due to arthritis or pain. The website should alert users that a false-positive diagnosis could result if they have any of these conditions.

Answer: We have updated the website to include a Caution message that addresses this concern, alongside the existing Requirements:

Requirement: Video should be at least 10 seconds long (processing takes approximately 30 seconds). The video must be in .mp4 or .mov format, and the duration of finger tapping should be at least 10 seconds.

Caution: If the test subject has muscle weakness, joint deformity, joint stiffness, or is unable to perform the finger-tapping test due to arthritis or pain, it may lead to a misinterpretation as bradykinesia, even though the person does not have Parkinson’s disease (false positive).

The authors should be commended for their effort editing the manuscript for clarity. The following list includes the remaining errors which need to be corrected.

line 38 "features was" should be "features were"

line 57 "patient" should be "patients"

line 76 "forassessing" should be "for assessing"

line 93 "perfrom" should be "perform"

line 95 "Four neurologists were rate 100 videos" should be "Four neurologists rated 100 videos"

line 133 "implementating" should be "implementing"

line 136 "stablizied" should be "stabilized"

line 142 insert a space before the bracket

line 177 insert a space after the comma

line 178 insert a space before parenthesis

line 179 "enviroment" should be "environment"

line 160 The text should be "within a 10-s window"

line 196 "for train the model" should be "for training the model"

line 197 "tune" should be "tuning"

line 214 insert a space before the bracket

line 222 "fail" should be "failure"

line 228 insert a space before the parenthesis

line 231 "slightmild" should be either "slight" or "mild"

line 254 "choice for for this" should be "choice for this"

line 257 replace "for the" with "to"

line 266 insert a space after "values"

line 281 insert a space after the period

line 285 insert a space before "[12]"

line 295 insert a space after the period

line 312 "classficiation" should be "classification"

line 319 "are various of" should be "are various"

line 329 "are" should be "is"

line 330 "it is can utilize" should be "it can utilize"

line 349 "tremors amplitude are" should be "tremor amplitude is"

line 352 insert a space after the period

line 366 "supportedby" should be "supported by"

In general, there should be a space before all brackets and parentheses, and there should be a space after all commas.

Answer: We have checked the grammar and typos.

---

## [Decision Letter · Decision Letter 3]

31 Jul 2025

Screening for Parkinson’s disease using “computer vision”

PONE-D-24-40785R3

Dear Dr. Panitchote,

We’re pleased to inform you that your manuscript has been judged scientifically suitable for publication and will be formally accepted for publication once it meets all outstanding technical requirements.

Kind regards,

John A. Thompson

Academic Editor

PLOS ONE

Additional Editor Comments (optional):

Reviewers' comments:

Reviewer's Responses to Questions

**Comments to the Author**

Reviewer #2: All comments have been addressed

2. Is the manuscript technically sound, and do the data support the conclusions?

Reviewer #2: Yes

3. Has the statistical analysis been performed appropriately and rigorously?

Reviewer #2: Yes

4. Have the authors made all data underlying the findings in their manuscript fully available?

Reviewer #2: Yes

5. Is the manuscript presented in an intelligible fashion and written in standard English?

Reviewer #2: Yes

Reviewer #2: (No Response)

**Do you want your identity to be public for this peer review?** For information about this choice, including consent withdrawal, please see our Privacy Policy

Reviewer #2: **Yes: ** W. Ryan Williamson, PhD

---

## [Editor Report · Acceptance letter]

PONE-D-24-40785R3

PLOS ONE

Dear Dr. Panitchote,

I'm pleased to inform you that your manuscript has been deemed suitable for publication in PLOS ONE. Congratulations! Your manuscript is now being handed over to our production team.

Kind regards,

on behalf of

Dr. John A. Thompson

Academic Editor

PLOS ONE